# Raman Spectroscopy as a Research and Diagnostic Tool in Clinical Hematology and Hematooncology

**DOI:** 10.3390/ijms25063376

**Published:** 2024-03-16

**Authors:** Paulina Laskowska, Piotr Mrowka, Eliza Glodkowska-Mrowka

**Affiliations:** 1Department of Experimental Hematology, Institute of Hematology and Transfusion Medicine, 02-776 Warsaw, Poland; plaskowska@ihit.waw.pl; 2Department of Biophysics, Physiology and Pathophysiology, Medical University of Warsaw, 02-004 Warsaw, Poland; 3Department of Laboratory Diagnostics and Clinical Immunology of Developmental Age, Medical University of Warsaw, 02-004 Warsaw, Poland; 4Department of Immunohematology, Institute of Hematology and Transfusion Medicine, 02-776 Warsaw, Poland

**Keywords:** Raman spectroscopy, Raman effect, diagnostics, hematologic malignancy, leukemia, lymphoma

## Abstract

Raman spectroscopy is a molecular spectroscopic technique able to provide detailed information about the chemical structure, phase, crystallinity, and molecular interactions of virtually any analyzed sample. Although its medical applications have been studied for several decades, only recent advances in microscopy, lasers, detectors, and better understanding of the principles of the Raman effect have successfully expanded its applicability to clinical settings. The promise of a rapid, label-free diagnostic method able to evaluate the metabolic status of a cell in vivo makes Raman spectroscopy particularly attractive for hematology and oncology. Here, we review widely studied hematological applications of Raman spectroscopy such as leukocyte activation status, evaluation of treatment response, and differentiation between cancer and non-malignant cells, as well as its use in still unexplored areas in hematology. We also discuss limitations and challenges faced by Raman spectroscopy-based diagnostics as well as recent advances and modifications of the method aimed to increase its applicability to clinical hematooncology.

## 1. Introduction

Hematology diagnostics is a complex, interdisciplinary field requiring a combination of various expertise and utilization of a vast array of diagnostic tools and methods [1,2]. This complexity is even more demanding due to the unique characteristics of hematologic entities such as acute leukemia, that require rapid diagnostic tools to allow for accurate and timely clinical decisions [3]. The success of immunotherapy in hematologic malignancies presented new diagnostic challenges, including the need for immune status evaluation. Although the recent introduction of high-throughput genetics has revolutionized this field, there are still many challenges to face and the development of new technologies able to overcome these obstacles, such as lack of accessibility, long time to diagnosis, and high expertise required, is still warranted [4].

Spectroscopic techniques are non-invasive methods to comprehensively investigate molecular systems in different environments and conditions. Medical applications of Raman spectroscopy, a molecular spectroscopic technique able to provide detailed information about the chemical structure, phase, crystallinity, and molecular interactions of a sample, have been studied for several decades. However, only recent advances in microscopy, lasers, detectors, and better understanding of the principles of the Raman effect have successfully expanded the application of Raman spectroscopy into clinical medicine [5]. So far, medical applications of Raman spectroscopy have been studied in more than a hundred clinical trials registered on clinicaltrial.gov database, both interventional and observatory ones. The promise of a rapid, label-free diagnostic method able to evaluate the metabolic status of a cell in vivo makes Raman spectroscopy particularly attractive for hematology and oncology [6,7]. Therefore, we decided to provide a comprehensive review of the advances of Raman spectroscopy in clinical hematology.

## 2. Raman Spectroscopy as a Research Tool in Medicine

### 2.1. General Principles of Raman Spectroscopy

Raman spectroscopy (RS) stands out among molecular spectroscopy techniques as it enables label-free and non-destructive detection and identification of molecules. This is achieved through the analysis of characteristic spectra unique to each molecular system [8]. RS operates on the principle of inelastic light scattering by matter, which was discovered by Chandrasekhara Venkata Raman in 1928 and named after him as the “Raman effect” [9]. When monochromatic light interacts with a molecule, most photons undergo elastic scattering, maintaining their energy and retaining the same frequency as the incident light (known as Rayleigh scattering). In contrast, a small fraction of the light experiences inelastic interactions with molecular vibrations [9]. This results in scattering at distinct optical frequencies determined by the energy gain (Stokes shift) or loss (anti-Stokes shift) (Figure 1). This subtle change in energy corresponds to the molecular vibrations associated with the chemical bonds within the sample, providing an insight into its molecular structure and composition. Analyzing how the distribution of inelastic scattering light intensity correlates with frequency allows for the differentiation of unique spectral patterns associated with various samples. The spectra produced by nucleic acids, proteins, lipids, and carbohydrates containing Raman-active functional groups can be employed to assess the metabolic state of different cells and tissue types, each characterized by its unique composition [10].

A Raman spectrum is generated by graphing the intensity of scattered light (for convenience often denoted in arbitrary units a.u.) against frequency (Figure 1). Typically, the frequency of scattered light is converted to Raman shifts, which represent the difference in energy between the original beam of light and the scattered light, usually measured in wavenumbers. Wavenumber is a unit that is defined as the number of wavelengths per unit distance, typically reciprocal centimeters (cm^−1^) but can be interchangeable with the wavelength expressed in nm [10]. The Raman shift is employed to ensure that data can be easily compared, even when different laser wavelengths are utilized.

Most chemical compounds exhibit Raman spectra in the fingerprint region (400–1800 cm^−1^), with some organic materials doing so in the 2700–3600 cm^−1^ high-wavelength region. There is also a silent region between 1800 and 2700 cm^−1^, where most of biological molecules do not produce Raman scattering [11]. The peaks within the Raman shift are characteristic for specific chemical groups or bonds. For example, a peak at 1002 cm^−1^ is a known wavelength of phenylalanine while 931 cm^−1^ corresponds to the left-handed helix DNA (Z form). Chemical specificity of Raman spectroscopy allows researchers to identify and characterize various cellular components and provide insight into the metabolic status of the cell, making it particularly interesting as a tool for clinical diagnostics.

### 2.2. Visualization and Analysis of Raman Spectra

Data obtained by RS can be presented as Raman spectra (wavelengths and numerical values), or Raman maps (images), the latter being visual mappings of Raman scores that present the distribution of the biochemicals within the studied sample (Figure 2A). Raman images reveal subcellular localization of studied substances and can provide quantitative data, incorporating statistics. Such maps can be correlated with optical or fluorescent microscopy images [12]. Raman spectra from an imaging experiment can be analyzed to obtain 1D profiles, 2D images or 3D rendered volumes.

Translation of Raman data into biologically and/or medically useful information requires utilization of complex statistical analysis and machine learning, of which a simplified scheme has been presented in Figure 2B. Most commonly used tools include principal component analysis (PCA), linear discriminant analysis (LDA) or supervised partial least-squares discriminant Analysis (PLSDA) [13]. 

The aim of PCA is to identify the components capturing the most variation in the unlabeled dataset while LDA is a supervised method and, as such, it operates on labeled data, assuming their normal distribution, when finding directions of maximum variance [8]. Both PLSDA and LDA are used in supervised machine learning to address multi-class classification problems, but PLSDA requires much more parameter optimization before reaching reliable and valid outcomes. PLSDA is especially popular in Raman analysis as spectral data have high dimensionality and the variables such as wavenumbers tend to correlate with each other [13]. 

Multivariate methods are used to reduce the data and to generate “scores” that summarize the main variation within the spectral data [14,15]. Alternatively, artificial neural networks are used for classification while regression analysis is used for extracting diagnostically significant peaks and spectral regions. Combining these techniques is common to enhance the accuracy and robustness of algorithms. To facilitate data analysis and further improve accuracy, preprocessing methods like filtering and evaluation approaches including k-fold cross-validation are frequently applied [15].

Machine learning (ML) techniques can further improve the classification and identification of Raman spectra. K-nearest neighbor (KNN) is the most basic ML non-linear classifier requiring no training process. It maps the samples to n-dimensional space upon calculating the distance input data using a distance metric formula. The support vector machine (SVM) is known as one of the best classification algorithms. SVM requires a training dataset to computationally determine the partitioning hyperplane for samples separation and by selecting the appropriate support vectors it categorizes the dataset into different classes [16]. The decision tree (DT) algorithm, employs a tree structure, simulating the human decision-making process. DT handles unrelated features within the dataset, but it can be overfitted. This issue is resolved in the more complex random forest (RF) model that incorporated multiple DTs where variables are randomly selected. RF enables the identification of important groups and variables within the dataset, it also enables to capture a deeper understanding of relations between variables [17]. Deep learning, ML branch based on neural network models, allows to omit the data pre-processing and shows highly promising learning capabilities and low generalization error compared with standard ML methods [16]. The most basic, artificial neural networks (ANNs), emulate the learning of the human brain adjusting and changing the connection weights of neurons, processing information and simulating the relationship between inputs and outputs through the training process [18]. Derivatives of ANNs have been developed, such as probabilistic neural network (PNN), recurrent neural network (RNN), convolutional neural network (CNN), and generative adversarial network (GAN). These networks are employed in modeling intricate non-linear systems, facilitating tasks such as clustering regression, classification, and prediction using Raman spectra datasets [16].

### 2.3. Applicability of Raman Spectroscopy to Medical Applications

Quantitative assessment of the biochemical composition of biological tissues using Raman spectroscopy makes it a valuable tool for medical applications. Moreover, the unique properties such as the potential for label-free, non-invasive in vivo diagnostics and monitoring at low cost and high speed make this technology even more promising. From technical point of view the method is very versatile, allowing to study different sample types, from body fluids, blood plasma or serum to cells and tissues. Giving up the cells in favor of fluids, provides easier material to work with and eliminates the noise due to unspecific signal from other cellular fractions; however, data obtained in this way lack the spatial distribution of biochemicals in the sampled tissue.

On the other hand, Raman spectroscopy has several limitations regarding its potential use in the clinic, resulting partly from the specificity of the Raman effect and partly from the limitations of optical methods. The Raman effect generates the signal of low intensity and thus the spontaneous Raman spectroscopy is time-consuming. The method, however, is constantly improved and several modifications of Raman spectroscopy are being developed to answer the challenges of modern diagnostics (summarized in Table 1). In the modified Raman microscopes, the problems with weak signal or background interference were addressed so the acquisition of spectra was accelerated, and background noise suppressed. Some other variations in Raman technique allow for quantitative measurements (stimulated Raman spectroscopy—SRS) or obtaining high-resolution maps with spatial distribution (Raman hyperspectral imaging). Also, some other optical methods are often combined with Raman to obtain new type of data that in correlation with Raman spectra would enrich the analysis [19,20]. 

## 3. Applications of Raman Spectroscopy in Medical Research and Clinical Studies

### 3.1. Raman Spectroscopy in Healthy Hematopoietic Cells

Studying normal hematopoietic stem/progenitor cells and their progeny presents several challenges, including limited sample availability and difficulties in maintaining their native status and function. Due to the complex activity of these cell populations, functional assays, allowing to determine cell activity, are often crucial for comprehensive evaluation of hematopoietic cells. However, experiments on living and intact blood cells without significant interference through labeling or other chemical treatment pose an additional technical challenge [32]. The unique properties of Raman spectroscopy, including single-cell resolution and label-free protocols, offer significant promise for understanding stem cell and mature cell populations heterogeneity, as well as for evaluation of cell development stages and activation status [33]. 

#### 3.1.1. Hematopoietic Stem Cells

Due to distinctive metabolic status, various subpopulations of stem cells are considered particularly good targets of Raman spectroscopic studies [34]. However, to the best of our knowledge, only a few initial attempts to study various populations of stem cells have been reported so far [35,36,37]. In the hematopoietic stem cell (HSC) field, initial experiments aiming to distinguish closely related hematopoietic stem and progenitor cell populations focused on surface-enhanced Raman spectroscopy requiring additional labeling [38]. Illin Y et al. [39] and Pastrana-Otero I et al. [40] have used partial least-squares discriminant analysis (PLS-DA) of RS to distinguish between closely related hematopoietic stem and progenitor cell subpopulations and predict their fate. Their recent approach allowed for location-specific identification of 6 murine hematopoietic stem and progenitor cell populations, i.e., HSC, multipotent progenitor (MPP)-1, MPP-2, MPP-3, common myeloid progenitor, and common lymphoid progenitor with high accuracy (90%) [40]. The same group was able to trace neutrophilic differentiation from hematopoietic progenitor cell to mature neutrophil with high specificity and sensitivity [41]. This non-invasive, label-free approach enables studying intact HSCs and progenitor cells and study their differentiation and fate decisions under various conditions.

Fluorescence-activated cell sorting (FACS) and magnetic bead cell separation, both based on cell surface marker immunostaining, are a mainstay of the subfractionation of hematopoietic cells for downstream analyses or functional studies. Similarly, Raman-activated cell sorting (RACS) is a label-free cell separation method utilizing differences in unique fingerprint characteristics of Raman spectra between cell populations instead of cell surface marker immunostaining to distinguish populations of interest [42]. Although the method can be used for eucaryotic cell separation, the most successful applications reported so far focus on microbial cell sorting [43]. Growing knowledge on Raman characteristics of HSC populations may soon enables using RACS for the label-free subfractionation of HSCs.

#### 3.1.2. Lymphocytes

Lymphocyte activation can be evaluated in different ways which are either indirect (ELISA of cytokine release) or require additional steps such as cellular staining (flow cytometry) or cell lysis (mRNA analysis). These methods are laborious and expensive, and in most cases require fixation or destruction of the studied cells making Raman spectroscopy an interesting alternative. Numerous groups reported successful discrimination of T cells, B cells and monocytes from both cell lines or PBMC samples using RS [32,44,45,46]. A single group achieved differentiation of five leukocyte subpopulations: monocytes, granulocytes (mainly neutrophils), T, B and NK cells using Raman microscopy [47], while others were able to separate lymphocytes, monocytes, neutrophils and eosinophils [48]. Other authors reported spectral identification of T and B cell activation status [31,49,50,51]. Brown et al. identified Raman spectra of late and early T cell activation, which was correlated with simultaneous mAb staining for surface markers [51].

Chaudhary et al. activated T cells, B cells and monocytes in vitro using stimulants, i.e., phytohemagglutinin (PHA), interleukin 4 (IL-4), and phorbol 12-myristate 13-acetate (PMA), respectively. They compared the spectra of activated vs. resting cells and built the classification models based on the spectral differences with high sensitivity (>90%) and specificity (>80%) [45]. Pavillon and Smith showed the RS-based statistical models discriminating activated and resting murine T cells, applying non-linear projection methods to illustrate changes occurring over several days during early differentiation. These results correlate with established surface markers of activation and differentiation [52]. Chen et al. used Wavelength Modulated Raman Spectroscopy (WMRS) for spectral identification of CD4+ T cells, CD8+ T cells and CD56+ NK cells subsets achieving specificities of up to 96%. In the same study, they were able to distinguish between CD303+ plasmacytoid and CD1c+ myeloid dendritic cell populations, yet with lower accuracy (specificity 87%, sensitivity 71%) [30].

Some researchers investigate the potential of RS in monitoring the differentiation and maturation of T and B cells. In 2021, Morrish et al. evaluated chromatin and transcriptional alterations during B lymphocyte maturation by employing confocal Raman microscopy along with correlative transcriptomics. This approach linked the variations in chemical and structural properties (mostly nucleic acids and proteins) to biological outcomes. The study involved the examination of live B cells before and after maturation. By identifying spectral variances between non-activated and activated B cells, the researchers assessed their correlation with known intracellular biological changes, comparing the results to RNA-seq analysis [53].

RS was also employed to identify glioma-associated neuroinflammation and distinguish T cells and monocytes after exposure to tumor-conditioned media from glioblastoma stem cells under various conditions. The treatment had an impact on the phenotype of T cells and monocytes, directing their differentiation into a mixed population displaying both pro-inflammatory and anti-inflammatory characteristics, which was confirmed with FC. These changes were reflected in the RS spectra differing in bands corresponding to lipids, proteins and nucleic acids from the untreated cells [54]. These findings show the utility of Raman spectroscopy in the inflammation monitoring.

In HemoSpec clinical trial (DRKS0000626), leukocytes from patients’ peripheral blood were measured with Raman spectroscopy and compared with the reference diagnostics (clinical scores, blood count, biomarkers—C-reactive protein, procalcitonin, and interleukin-6). The information extracted from the Raman spectra reflects the distinct chemical profiles of leukocytes from patients with inflammation, infection, and sepsis. Based on RS data, classifications models were built to distinguish patients with and without infection as well as with and without sepsis. These findings require additional investigation as they hold the potential for application in patient stratification [55].

#### 3.1.3. Monocytes and Macrophages

Monocytes seem to be particularly good object of Raman studies due to their distinctive morphology compared to the other blood cells. LPS-stimulation mimicking TLR4 activation of THP-1 monocytic cell line was identified by RS. The changes induced by LPS in DNA, lipids, and protein structure and composition resulted in a distinctive spectral fingerprint for TLR4 activation. Specific and characteristic spectral features were identified in the Raman spectra of several individual cells at all the different time points following LPS stimulation [56].

Macrophages, derived from monocytes, can be broadly categorized into three groups, based on their function and phenotype: resting M0 cells, pro-inflammatory M1 cells and anti-inflammatory M2 cells [57]. Ribeiro et al. polarized the murine model of macrophages J774A into M0, M1 and M2 phenotypes and compared their spectral profiles. Despite the subtle spectral differences observed, the multivariate model employed quantified the separation in the PCA scores plot for M1 and M2 spectra, showing a statistically significant separation between the clusters [58]. Similar results were observed in the study by Nauman et al. which showed discrimination models distinguishing between M1 and M0/M2 macrophages with an accuracy of 85–90% built on RS data obtained from healthy donors. Raman images were analyzed using the spectral unmixing algorithm N-FINDR to visualize the chemical distribution within cells. The resulting Raman maps strongly resembled fluorescent and confocal laser scanning microscopy images of M0, M1 and M2 macrophages taken from the same samples [12]. It shows both utility of chemometric-based classification models and that Raman maps can be a label-free alternative to imaging requiring additional staining.

Feurer et al. characterized Raman spectra of human-derived macrophages, polarized into four subgroups—M0, M1, M2a (IL-4/IL-13-stimulated) and M2c (IL-10)—but focused mostly on lipid fingerprint region [59]. The significance of lipid differences in RS spectra corresponds with results of previous studies, which reported that fatty acid composition is strongly related to macrophage metabolism and activation [60]. In this study, macrophage response to titanium (Ti)—material mostly used for transplants—was additionally investigated with RS. Standard methods, such as surface antigen analysis or cytokine expression, showed no surface-induced changes in macrophages while RS detected changes in lipid composition and biochemistry of tested cells. Ti-plated macrophages resembled M2 while standard glass-plated macrophages did not share characteristics either with M1 or M2 [59]. Findings reflect an increase in anti-inflammatory phenotypes of macrophages cultured on Ti in vitro [61].

#### 3.1.4. Future Perspectives

Although Raman spectroscopy seems to be a very promising method to monitor activation and differentiation of hematopoietic stem cells and mature blood cells, further research whether it can be employed in more subtle subgroup identification (like Th1, Th2 and Th17 cells for CD4+ T lymphocytes) is still needed. There is a noticeable decrease in specificities of discrimination between some of studied subpopulations that may suggest the RS method has some limitations or it would benefit from coupling with different optical tools or different multivariate analysis. Raman spectroscopy applications in studies of healthy immune cells are summarized in Table 2.

### 3.2. Raman Spectroscopy in Tumor Studies

Malignant transformation requires multiple genetic and metabolic changes that confer the ability to grow uncontrollably and spread. These changes are difficult to detect at early stages making timely and accurate cancer diagnosis, that is crucial for therapeutic success, challenging. In the last decade cancer diagnostics has evolved rapidly, revolutionizing clinical oncology by improved effectiveness of cancer screening, better understanding of tumor biology and enabling personalized treatments that translate into better patient outcomes. Hematooncology, focusing on malignant diseases originating from hematopoietic stem and progenitor cells, is a distinct branch of oncology with unique specifics, including fulminant course of some acute entities unseen in other medical fields, non-solid characteristics of tumors, and complex relationship with the immune system, as the immune cells are both the pursuer and the pursued. Although it benefited greatly from the recent progress, it came at the cost of development of interdisciplinary and complex diagnostic protocols that are time and resource consuming. Therefore, there is still much room for improvement and the development of a label-free, fast, and cheap method for identification of malignant cells that could become the mainstay of hematologic malignancy diagnostics would be most welcome. 

#### 3.2.1. Cancer Screening

The main challenge in the development of effective cancer screening methods is to distinguish healthy cells from malignant cells with high sensitivity and specificity at early stages of development. As the access to tumor cells at these stages is very limited and the number of transformed cells is usually low, many researchers decide to use indirect screening looking for metabolic changes in serum or blood plasma rather than tumor cells themselves.

In the study from 2022, Woods at al. presented the assessment of Raman spectroscopy in rapid diagnostic from the serum collected from patients with vague cancer-related symptoms. In the 54 collected samples, 10 patients were found to have cancer. For these samples, three peaks were identified to be significant in comparison to the spectra of healthy control via statistical analysis. These changes, attributed to proteins, amino acids, lipids, fatty acids, glycoproteins, carbohydrates, and carotenoids, were linked to hallmarks of cancer, to the metabolic changes and energetical adaptations ongoing in the cancerous cells such as glutaminolysis, aerobic glycolysis, and pathological mitochondrial alteration [62]. These markers could potentially be used for preliminary testing of patients for changes which could be indicative of cancer presence. Although the study referred to various solid tumors, basic principles, allowing to distinguish cancer cells from non-malignant tissue provide a proof- of concept for a similar approach in hematooncology.

Another interesting approach was to use peripheral blood cells as a marker of tumor immune response, an indirect proof of the presence of malignancy in the system. As natural killer cells (NK cells) are the first line of defense against cancer and demonstrate preferential cytotoxicity towards cancer stem cells (CSCs) [63], the presence of CSCs was determined by observing changes in NK cell expressions. NK cells were cocultured with cancer cells, CSCs and non-cancer cells and collected spectral signals using the SERS functionalized OncoImmune Probe Platform. Cancer-associated NK spectra were distinctive from naïve and CSC associated and were used for a cancer-prediction model, which was tested on 22 patient samples. As little as 5 uL of peripheral blood was sufficient to obtain 100% accuracy in distinguishing cancer from non-cancer sample and correct localization (breast, colorectal of lung) of cancer with 93% accuracy based on spectral features of NK cells. These results show the potential of minimally invasive cancer screening using circulating NK cells [64].

#### 3.2.2. Hematopoietic Malignancy Diagnostics and Subtyping

In most cases, hematologic malignancies, due to their non-solid nature, are easily accessible through blood or bone marrow sampling. Therefore, direct methods of cancer cell identification, requiring differentiation between healthy blood or bone marrow cells and malignant cells, are the most effective. Early attempts to discriminate between healthy and malignant cells using Raman spectroscopy were performed on cell lines. Healthy B cells and B-cell acute lymphoblastic leukemia (ALL) cell line MN60 were discriminated based on Raman spectra [47]. Also, the feasibility of discrimination between subtypes of hematologic malignancies (lymphoma) was shown in cell line models. For example, the comparison between non-Hodgkin lymphoma (NHL) JMP-1/MCL and Hodgkin lymphoma MDA-V cell lines allowed for identification under different temperature as the measurements were taken in the range between 15 and 37 °C. They demonstrated that temperature influences specific Raman peaks in the nuclei of the cell line, although it does not contribute to the differentiation of the cell lines [65]. It proves that RS is reliable method despite a potential physicochemical changes (temperature, humidity) that could interfere with the analysis.

Hassoun et al. showed high sensitivity method for acute myeloid leukemia (AML) cells detection. Raman spectra of lysates from healthy monocytes and THP-1 (AML cell line) cell line were identified, then lysates were mixed in different proportions and RS was performed again on the samples. Then, they used non-negative least square (NNLS) algorithm enabling the calculation of percentile amounts for each cell type and determination of their contributions to the mixtures. The study underscores the potential of SERS with NNLS fitting as a rapid method for diagnosing AML in human blood or BM samples [66]. Similar results were obtained by Yu et al. who discriminated human myeloid leukemia cells (HL60 and K562) from healthy bone marrow samples with SERS. Overall accuracy of model built on Raman spectra exceeded 96% [67].

Da Silva et al. compared Raman spectra of plasma samples and whole blood from AML patients. The spectral differences in plasma samples were related to carotenoids, while in whole-blood samples, they were primarily associated with amino acids and proteins. Based on spectral data, two sets of models to discriminate healthy from leukemic spectra were built. For the one based on plasma data supervised partial least-squares regression (PLS) accuracy was 97%, unsupervised principal component analysis (PCA) it was 64%, while for the models based on whole-blood spectra PLS accuracy was 96% and PCA accuracy was 93%. These findings support the possibility of AML diagnostic use of Raman spectroscopy on both plasma and whole-blood samples [68].

The potential of SERS was also studied in developing a simple blood test for non-invasive diffuse large B-cell lymphoma (DLBCL) and chronic lymphocytic leukemia (CLL) differentiation. The predictive models built on RS spectra from plasma, differentiates DLBCLs/healthy or CLL/DLBCL or CLL/healthy populations. The authors suggested that the combination of Raman shifts at 1445 cm^−1^ and 1655 cm^−1^ is distinctive for DLBCL, allowing differentiation from CLL and even from other solid tumors documented to date [69].

Raman-enhanced spectroscopy (RESpect) probe advances RS technology with a portable fiber-optic device. To assess its clinical potential, RESpect was used to analyze spectra of different pediatric NHL tissues and non-malignant specimens and compare them with standard RS. Fingerprint comparisons between standard Raman spectroscopy and the RESpect probe demonstrated comparable primary peaks and ability to identify a spectrum of pediatric NHL and follicular hyperplasia (FH) [70]. It is another study showing that Raman technology could serve as an initial step in determining when a definitive tissue biopsy is necessary. In another study, spectra of children NHL cell lines (Ramos and CA46) were tested against healthy B cells proving the ability to distinguish them with accuracy and specificity of 100% [71].

In the DOLPHIN VIVO trial (NCT04162431), an FNA/Raman spectroscopy needle probe was designed to measure and gather data on excised lymph node tissue to detect lymphoma. Spectra from Raman and FTIR were correlated with routine histopathology results using multivariate analysis methods and compared to cytology samples. The ex vivo study will be followed by the in vivo trial that has not started recruiting patients yet. By combining RS with a fine needle probe, the technique can target tissues below the skin with minimal invasion. The needle probe should provide the clinician with instant diagnosis without the delay and cost of a laboratory analysis by pathologists [25]. 

Lymph nodes from 20 patients, who underwent surgery for suspected lymphoma, were examined collecting Raman maps and hundreds of spectra of obtained tissues. Using PLS-DA, lymph node classification models were generated, firstly to distinguish between benign (reactive FH) and malignant (follicular lymphoma FL and DLBCL) tissues and to further discriminate among cancer types and grades. Models showed accuracy higher than 88% in all cases, being validated on samples previously unused for models development [72]. In this study, the spectral differences regarding the expression of the BCL2 protein characteristic for FL and some cases of DLBCL [73] were identified, which led to developing a RS-based classification of FLs according to BCL2 expression. RS revealed significant differences between Grade I-II and Grade III follicular lymphomas, thus indicating that increasing histological malignancy is associated with distinct changes in the spectra [72]. 

Similar distinction was reported by Chen’s group that measured serum samples of DLBCL patients at different progressive stages and healthy controls using SERS. Chemometric analysis of spectra led not only to distinguishing between healthy and DLBCL spectra but also enabled building prediction models for discriminating early (I and II) and late (III and IV) stages of DLBCL with accuracy >90%. The serum-based SERS in combination with multivariate analysis could serve as a potential technique for non-invasive diagnosis and staging of DLBCL [74]. The other study performed on DLBCL cell lines allowed for building prediction models for discriminating malignant cells from healthy B cells. Next, models were trained to distinguish ABC vs. GCB and Ox-Phos vs. non-OxPhos subtypes within DLBCLs [75]. This could be a potential tool for classifying and recruiting DLBCL patients to the clinical trials based on their metabolic profile.

A cell line-based approach was also used in the study for classification of B-cell leukemia into distinct differentiation/maturation stages. RS4;11, REH, MN60 cell lines, each representing different differentiation subtypes of ALL, were measured with RS. In proposed PCA analysis, the three B-leukemia cell types and healthy B-cell counterpart formed four distinct groups, which was the basis for creating a predictive model for their distinction (97% efficiency). The models were then tested on two patient samples. Despite higher spectral variability in the clinical samples compared to the cell lines, RS analysis demonstrated the discrimination of B-ALL clinical samples with high sensitivity (>88%) and specificity (>85%) compared to normal B cells [76].

In different study, detailed Raman maps of AML patient bone marrow cells were evaluated [77]. At diagnosis, the samples were categorized with the French–American–British (FAB) classification into M0, M2, M3, and M6. This morphology-based classification does not overlap with the most recent WHO AML classification [73] based on a combination of genetic and immunophenotypic markers and morphology. Obtained RS spectra were used to calculate the average fingerprint spectra of each subtype and to compile Raman maps that showed localization of the cytoplasm, hemoglobin, nucleus, and granules with myeloperoxidase, reflecting the morphological differences of FAB subtypes. The multivariate analysis revealed at first a good distinction between erythroleukemic and myeloblasts groups, showing in PCA plot that promyelocytic M2 and M3 cells were mixing but based on the averaged intensity of DNA and myeloperoxidase peaks all four subgroups differed statistically. The model built in this study allowed for classification of M0 and M6 with 100% accuracy, specificity, and sensitivity. Promyelocytes (M2 and M3) were efficiently discriminated from two other groups with 100% accuracy. This type of RS analysis could potentially substitute the manual examination of bone marrow and peripheral blood smears of AML/myelodysplastic (MDS) patients [77].

Another attempt of RS-based AML/MDS rapid identification was performed by the Liang group on patient sera. Raman spectra revealed significantly lower intensities of collagen and carbohydrate peaks in the MDS compared to the AML group, potentially attributed to changes in the patients’ body metabolism. Nine validation models comparing different variations in AML and MDS were used to evaluate the RS-based multivariate model’s accuracy. Validation specificity ranged between 92% and 100% and sensitivity between 75 and 100% [78]. Serum-based Raman spectral analysis could be a rapid label-free identification tool that would improve the diagnostic efficiency of MDS and AML as MDS can develop into secondary AML and its diagnostics is mostly exclusive [79].

RS and surface-enhanced Raman spectroscopy (SERS) were used for exosome screening to discriminate between patients across the three different clinical conditions: monoclonal gammopathy of uncertain significance (MGUS)—the predecessor of multiple myeloma, asymptomatic MM (aMM) and symptomatic MM (sMM). Fine tuning of chemometric analysis enabled identifying specific peaks for all three groups of exosomes. Main differences between aMM and sMM exosomes were found in spectra characteristic for lipids and their saturation degree which is consistent with literature reporting that lipids have a huge role in exosome formation, composition, and the execution of their signaling roles [80].

In the study from Canada, nano-SERS was employed to monitor minimal residual disease in patients plasma samples. Spectra of four cohorts: healthy, cardiovascular (CVC) patients and hematological patients divided into MM and lymphoma were measured. Model to discriminate between the cancer patients and healthy subjects had 79% accuracy (similar healthy vs. sick—both patients with cancer and CVC). Yet, for a more challenging three-class problem, discrimination between CVC, cancer and healthy subjects, it was above 69%. Over the treatment period, nanoSERS spectra of cancer patients changed, enabling the construction of a predictive model for disease burden that correlated with clinical parameters. The most effective model, relying on 15 features, could estimate the disease burden with a mean square error of 1.6 [24]. It proves that nano-SERS could be a potential alternative to existing methods for monitoring treatment response and detection of minimal residue disease (MRD) which are complex, have low sensitivity or remain expensive [81]. Different approaches to RS application in hematooncology diagnostics are summarized in Table 3.

#### 3.2.3. Therapy Monitoring and Treatment Efficacy Evaluation

Due to the possibility to observe metabolic changes withing the cell using RS, it is a potential tool for label-free and rapid monitoring of drug sensitivity and response surrogates. We present few cases that take a spectral approach, combining it with statistical analysis for treatment assessment, so far only in vitro.

To mimic the clinical setting, B-ALL cell lines on different stages of differentiation were treated with low doses of two drugs beneficial in the maintenance chemotherapy of B-ALL patients—methotrexate (MTX) and 6-mercaptopurine (6MP) and all-trans-retinoic acid (ATRA) as a control. MTX- and 6MP-treated cells showed signs of suppressed differentiation while ATRA had no effect, which was reflected in Raman spectra changes compared to untreated cells. Multivariate analysis correctly distinguished MTX- and 6MP-treated cells from controls while spectra of ATRA-treated cells showed huge overlap with untreated cells. This confirms RS utility in monitoring therapy and suggest its potential in the identification of MRD [76].

In the study involving a large pool of spectra, where THP-1 cells were treated with doxorubicin (DOX), the alterations within Raman spectra of DOX-treated cells can predominantly be attributed to changes in nucleic acid, protein, and DOX content. These changes align with the biological effects of that chemotherapeutic drug on cells, involving drug–DNA intercalation, DNA denaturation and DNA replication stop. Differences in protein and lipid content reflect apoptotic phenotype of cells caused by DOX treatment. In this study, it was also demonstrated that changes in spectra were dose-dependent and that they could serve as a Raman-based viability assay. The high-content analysis Raman spectroscopy (HCA-RS) platform developed for this study enables the execution of an analytical series without human intervention. It also allows sampling of a statistically significant number of cells under various physiological conditions within mixed populations [26]

In the study from 2017, FTIR coupled with RS was used to monitor kinetics and intracellular mechanism of the redeployed drug combination of bezafibrate and medroxyprogesterone acetate (BaP) in HL-60 (AML) and K562 (chronic myeloid leukemia—CML) cells [19]. BaP has shown anti-leukemic activity in vitro and in vivo [84]. Raman maps showed no obvious differences between treated and untreated cells but supervised PLS-DA analysis revealed that BaP treatment induced three RS peaks originating from cellular lipid changes. Spatial distribution confirmed no differences between membrane lipids and intra-cellular lipids but rather consistent lipid changes across single treated cell spectra. RS enabled the tracking of the impact of drug therapy on the cells biochemistry, providing a deeper understanding of the targets of BaP [19].

SRS was applied to live monitoring of ponatinib treatment of CML cell lines [28]. RS spectra of treated KCL22 cells exhibited differences in proteins and had a distinctly elevated peak of alkyne which localizes in the silent region of Raman spectra (2221cm^−1^), due to the chemical composition of the drug—ponatinib has an alkyne moiety [85]. Comparison of Raman signal in the resistant KCL22^Pon-Res^ cells and parental KCL22 cells revealed that mean the intensity of ponatinib increased 1.9 fold in the resistant cells compared to parental KCL22 cells. Raman imaging confirmed the entrapment of the majority of ponatinib inside acidic organelles, most likely lysosomes. After pre-treatment with lysosomotropic agent—chloroquine, Raman signal of ponatinib was at least 3-fold weaker than in previous experiment but BCR-ABL inhibition increased. The application of SRS microscopy enabled label-free imaging of the ponatinib and provided insights into changes in the uptake of the drug that occurred during the development of acquired drug resistance. The results suggest that by adding tags that excite scattering in silent region SRS imaging of drugs would have increased sensitivity, extending the potential for this technology to provide read-outs of drug kinetics and mechanism of action [28].

The chemotherapy treatment at different concentration revealed a gradual decrease in the intensities of Raman peaks associated with nucleic acids and proteins. The decrease was proportional to the increase in methotrexate (MTX) concentrations, ranging from 0.01 to 1 μM. These spectral changes have been previously linked to the mechanisms of action of MTX [86]. With a prediction accuracy exceeding 90%, the model suggests that Raman microscopy can effectively track and monitor the spectral modifications induced in leukemia cells during chemotherapy treatment [47].

In another study, confocal micro-RS was used for distinction between dexamethasone- or bortezomib-resistant and sensitive MM cell lines. Distinguishing features, primarily arising from variations in DNA/RNA ratio, nucleic acids, lipids, and protein concentrations, allowed for discerning the sensitive and resistant subtypes [82]. The identification of drug-resistant cells by RS could be a clinical tool to assess the development of resistance to glucocorticoids and proteasome inhibitors in myeloma cells. A study from Kang et al. in 2016 reports, real-time near infrared (NIR) RS could be used to monitor the proteasome inhibitors treatment on MM cells. Cells exposed to bortezomib showed weaker bands of nucleic acids, which can be associated with DNA damage, and elevated peaks of several amino acids corresponding to the accumulation of proteins due to the proteasome inhibition [83]. For this kind of analysis RS was also paired with another optical method—digital holographic microcopy that provides more physical information about the cells [20]. The higher density in the nuclear region after bortezomib treatment matching the elevated protein Raman bands, further confirmed the utility of optical methods in monitoring proteasome inhibition.

## 4. Conclusions and Perspectives

In most studies, Raman spectroscopy is described as a promising method that could become a reliable diagnostic tool. It has many sought-after properties as it is non-invasive, label free, requires only a small sample volume/size, enables real-time analysis, and provides metabolic information without destruction of the cell. RS-based predictive models are characterized with high specificity and sensitivity, demonstrating low risk of false positive and false negative results if adapted for clinical use. Spectral predictive models could be a first line of screening diagnostics, providing fast and non-invasive assessment of patient’s status. Yet, despite almost one hundred years of development, the technique has not been widely implemented into the clinic.

The reason for this may be that modern Raman spectroscopy is all but unified and standardized. Many researchers build their imaging platforms from scratch or adapt the commercially available Raman microscopes for the specific task, making the results difficult to compare between research groups and projects. The multitude of RS modifications provides an opportunity to bypass the limitations of the method itself but does not allow for the data reproducibility or comparison between datasets obtained in different studies. Although data from SRS, RS or SERS show some universal trends, data acquisition and processing need to be adjusted to each system. Consequently, research protocols are not unified, nor are pipelines of machine learning algorithms. There is also a reported problem with reproducibility of measurements in the same settings but there are few published attempts proposing standards of good practices to improve it [87,88].

Note that data acquisition quality standards also vary significantly between groups and studies. The number of spectra acquired is often relatively small, varying from tens [71] to a few hundred [78] with very rare cases where number of measured spectra exceed a few thousands [26]. While RS technology emphasizes the single-cell approach, the efficacy and reliability of the statistical analyses that are performed on gathered Raman spectra increase with the volume of data involved. Therefore, a higher number of spectral data should be obtained for more complex analyses to improve their statistical power and quality. However, there are no public repositories of human cell RS spectra (there are such for materials), so each research project requires preparation of new spectra dataset tailored for the tested hypothesis. 

The heterogeneity of the studied groups and their limited size pose further limitations. Researchers often admit in the discussion sections that studies would benefit from more comparable groups in terms of age, sex and even number of samples as the disproportions may affect the results. Consequently, lack of comprehensive studies performed in large patient cohorts delays progress in application of the Raman spectroscopy in the clinic.

In addition to technical issues described above, there are several challenges that are inherent to the field of hematooncology. The WHO classification of hematological malignancies is heavily based on genetics [73]; therefore, developing the RS method towards identifying individual mutations would increase the utility of spectroscopy-based analysis in hematological diagnostics. To achieve that, some researchers correlate their Raman data with transcriptomics and genetic information from patient sample. There are a few reports of attempts to identify hematological malignancies based on single mutations [89,90]. The feasibility of this approach has been reported previously for solid tumors [91,92,93] including mutations typically observed in leukemias, e.g., IDH [93,94].

There are still unexplored areas in hematology, where Raman spectroscopy could prove useful. As RS has been proven to accurately track immune cell states, it could be used for monitoring immunotherapies such as CAR-T or therapies involving hematopoietic stem cells [95]. Also, the field of hematopoietic stem cell research, including label-free identification and sorting of these cells, remains largely uninvestigated. RS could also be used in initial phases of clinical trials to monitor biochemistry and drug kinetics of newly proposed small-molecule drugs or monoclonal antibodies in various settings.

Hematooncology, as a rapidly evolving field, requires constant research and development of new diagnostic tools, allowing for accurate and timely clinical decisions that translate into therapeutic success. Raman spectroscopy, with its high sensitivity, ability to study intact cells, and real-time analysis capabilities, holds the potential to address many challenges of contemporary diagnostics. The future will show if this potential can be translated into clinically meaningful discoveries.

## Figures and Tables

**Figure 1 ijms-25-03376-f001:**
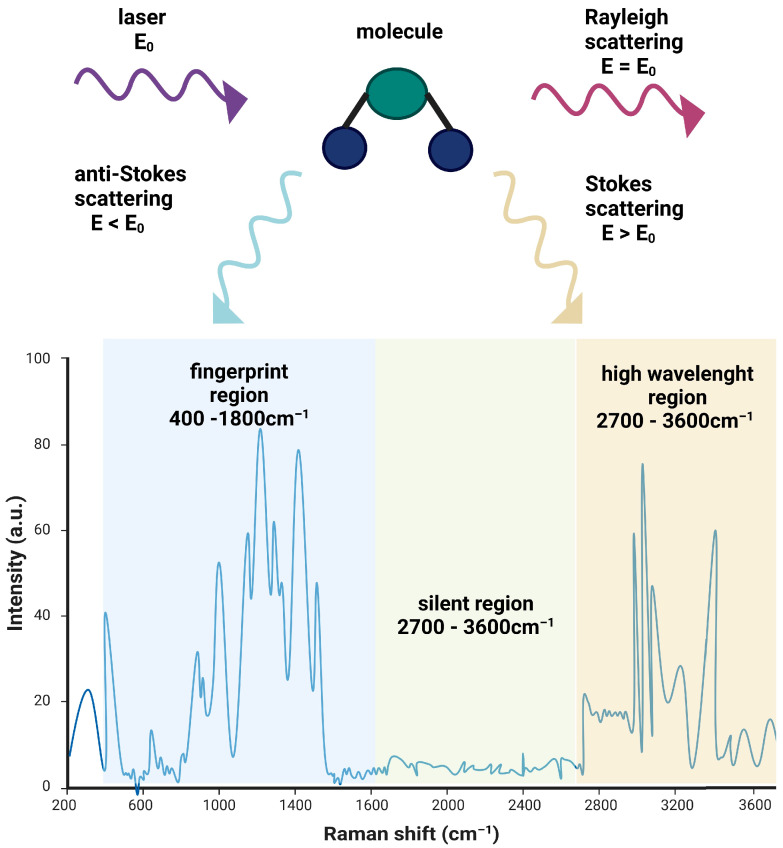
Schematic illustration of the Raman effect. When monochromatic radiation (laser, E_0_) interacts with a molecule, most of the incident radiation undergoes Rayleigh scattering (E = E_0_) but a small fraction of the scattered radiation exhibits a different frequency. This can manifest as lower-frequency anti-Stokes scattering (E < E_0_) or higher-frequency Stokes scattering (E > E_0_), collectively referred to as Raman scattering (**upper panel**). A Raman spectrum shows the intensity of scattered radiation as a function of the scattered photon’s energy, expressed as a change in the wavenumber, and can be divided into a fingerprint region, a silent region, and a high-wavelength region (**lower panel**). Created with BioRender.com.

**Figure 2 ijms-25-03376-f002:**
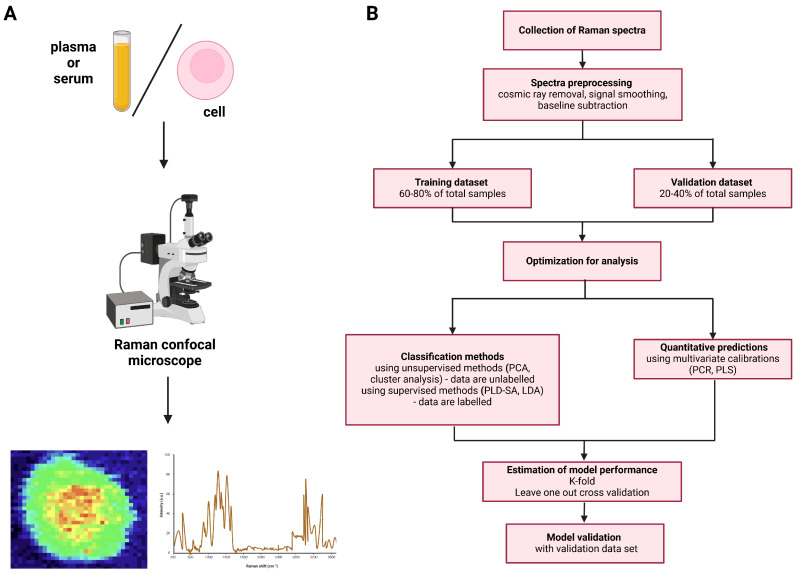
(**A**) Raman data acquisition and (**B**) data processing from the collection to prediction model validation. Created with BioRender.com.

**Table 1 ijms-25-03376-t001:** Modifications of Raman spectroscopy for medical application.

Technology/Modification	Raman Limitation Addressed	Principle	Use In Research/Diagnostics	Reference
Raman hyperspectral imaging (his)	In vivo tissue visualization with high resolution	Combines spectral information with spatial information to generate image showing distribution of biochemical components within the sample	Distinguishing of grey matter, white matter, GBM and necrosis within the sample	[21]
			Distinguishing between ovarian cancer and healthy plasma samples	[22]
Surfaced-enhanced Raman spectroscopy (SERS)	Low sensitivity	By addition of a metal (Au or Ag), enhances the Raman scattering signal of molecules close to the surface and eliminates fluorescent background	Establishing a predictive model to evaluate changes in the disease progression over time within patients with MM or lymphoma	[23,24]
Fourier- transformation infrared (FTIR) Raman spectroscopy		Measures the absorption, reflection, or transmission of electromagnetic radiation in mid-infrared complimenting chemical data obtained from RS	Vibrational spectroscopy signal characterization in head and neck lymph nodes (via Raman and FTIR mapping measurements of tissue sections)	[15,25]
High-content analysis Raman spectroscopy (HcA-RS)	Lack of automatization	Enables the sampling of a large number of cells under various physiological conditions without requiring user interaction	Spectral measurement of large number of samples—>25,000 spectra obtained for set of analysis	[26]
Stimulated Raman spectroscopy (SRS)	Slow acquisition	Provides much stronger signal which speeds the acquisition and eliminates a non-resonant background of spontaneous RS. With a linear relationship between signal intensity and chemical concentration, enables quantitative imaging	Real-time measurements of ponatinib distribution in live CML cells with high sensitivity and resolution	[27,28]
Wavelength-modulation Raman spectroscopy (WMRS)		suppresses the Raman background and speeds the acquisition time	Identification of T cells, NK cells and dendritic cells	[29,30]
Integrated Raman and angular-scattering microscopy (IRAM)	Lack of morphological information	Simultaneous measurements of elastic and inelastic scattering	Chemical and morphological distinction between activated and non-activated CD8+ T lymphocytes	[31]

Abbreviations: GBM glioblastoma, CML chronic myeloid leukemia, MM multiple myeloma, and NK natural killer.

**Table 2 ijms-25-03376-t002:** Summary of Raman spectroscopy application in studies of healthy immune cells.

Cell Population	Application	Raman Technique	Reference
lymphocytes	activation status assessment	RS, SRS, FTIR Raman, IRAM	[31,33,49,50,51]
	differentiation and maturation	RS	[52,53]
leukocytes	distinction between healthy and leukocytes with inflammation.	RS	[55]
monocytes and macrophages	activation status assessment	RS	[33,56,59]
	polarization	RS	[12,58,59]
HSC and progenitor cells	classification and fate prediction	RS	[39,40,41]

Abbreviations: RS spontaneous Raman spectroscopy, SRS stimulated Raman spectroscopy, FTIR Fourier-transformation infrared, IRAM integrated Raman and angular-scattering microscopy, and HSC hematopoietic stem cell.

**Table 3 ijms-25-03376-t003:** Summary of Raman spectroscopy application in hematological diagnostics.

Studied Sample	Application	Raman Technique	Reference
Serum	Solid tumor detection	RS	[62]
	DLBCL diagnosis and staging	SERS	[74]
Blood plasma	CLL screening	SERS	[69]
	DLBCL screening	SERS	[69]
	AML screening	RS	[68]
	MM screening	RS, SERS	[76,81]
Tumor-associated NK cells	Solid tumor detection	SERS	[64]
ALL cells	Distinction from healthy cells	RS	[47]
	Subtyping	RS	[76]
	Treatment monitoring	RS	[76]
NHL cells	Subtyping	RS	[65]
	Screening (diagnosis)	RS, RESpect (SERS), FTIR Raman	[25,70,71]
AML cells	Distinction from healthy cells	SERS	[66,67]
	Subtyping (based on FAB classification)	RS	[77]
	Differentiation from MDS	RS	[77,78]
	Treatment monitoring	HcA-RS, FTIR Raman	[19,26]
DLBCL	Subtype classification	RS, SERS	[75,80]
	Classification and staging	RS	[72]
MM	Drug resistance/ sensitivity studies	RS	[82]
	Treatment monitoring	RS, NIRS	[20,83]
	Diagnosis and MRD detection	RS, SERS	[76,81]
CML	Treatment monitoring	SRS	[28]

Abbreviations: RS spontaneous Raman spectroscopy, SRS stimulated Raman spectroscopy, NK natural killer, SERS surfaced-enhanced Raman spectroscopy, ALL acute lymphoid leukemia, NHL non-Hodgkin, FTIR Fourier-transformation infrared, AML acute myeloid leukemia, FAB French–American–British, MDS myelodysplastic syndrome, HcA-RS High-content analysis Raman spectroscopy, CLL chronic lymphocytic leukemia, DLBCL diffuse large B-cell lymphoma, MM multiple myeloma, NIRS near-infrared Raman spectroscopy, MRD minimal residue disease, and CML chronic myeloid leukemia.

## Data Availability

Not applicable.

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
