# Peer review of "Raman Spectroscopy as a Research and Diagnostic Tool in Clinical Hematology and Hematooncology"

_ijms, 2024, doi:10.3390/ijms25063376_

Round 1

Reviewer 1 Report

Comments and Suggestions for Authors

The manuscript is an interesting work that introduces a reader to the very basics of different Raman spectroscopy techniques and then reviews modern studies that applied Raman spectroscopy to hematology. The manuscript is well-structured and easy to read except for a couple of paragraphs (see below). My only concern is that several inaccuracies exist in the part that describes the physical basis of the Raman spectroscopy. It is the only reason I would recommend  a “major” revision” (but not “minor revision””). After correction of these inaccuracies, the manuscript will be a valuable source of information especially for novices in the field.

Corrections:

Line 81-85

“The intensity of inelastically scattered photons, plotted against the energy shift, can           81

be expressed in wavenumber values and represented as a Raman shift. The frequency of 82

scattered light is converted into Raman shifts, which indicates the difference in frequency  83

between the incident and scattered light. This conversion is typically denoted in units of       84

wavenumbers (cm⁻¹) (Fig. 1) and presented in a form of Raman spectrum [3]”.

This paragraph has to be carefully rewritten. It is just quite wrong. Intensity of light can not be expressed in wavenumbers. Wavenumber is a  unit that is defined as  the number of wavelengths per unit distance, typically reciprocal centimeters (cm−1) and it corresponds to the energy of a photon but  not directly to the intensity of the light beam . This unit can be used as interchangeable with the wavelength that, i.e., can be expressed  in nm. cm−1 is widely used in spectroscopy and not only for Raman spectra. The units for the intensity of light are “watts per square meter”, “watts per steradian” etc. In Fig. 1 a.u. are stated, and that means arbitrary units, i.e. it is much easier to measure relative intensity than the absolute value. The Raman shift is the difference in energy between  the incident light  and the scattered light; The Raman shift  can be expressed not only in cm−1 but for example in nm, although  in cm−1 is very convenient. 

Line 86 “Most chemical compounds emit Raman spectra in the fingerprint region”

It is more safe to say “exhibit” but not “emit”

Line 138

“Raman effect is a weak phenomenon and due to low 138

intensity of the signal, the spontaneous Raman spectroscopy is time-consuming”

It is better to rephrase – a phenomenon can not be weak, a signal can be weak.  

Also “spontaneous Raman scattering” is not defined before but is used. To avoid  reader’s confusion, the general term RS, which is used before, should be splitted into two: the “spontaneous Raman scattering” and “stimulated Raman scattering”, which is discussed below. 

Comments on the Quality of English Language

In total, the manuscript is written in easy-to-follow English.For some reason, abstract 12-24 and especially the paragraph 3.1 153-168 stands out from the rest and requires more editing to get to the level of clarity  achieved for the rest of the manuscript.

Author Response

The following corrections have been made:

Line 81-85

The paragraph 81-86 was rewritten according to the Reviewer’s corrections:

A Raman spectrum is generated by graphing the intensity of scattered light (for convenience often denoted in arbitrary units a.u.) against frequency (Fig. 1). Typically, the frequency of scattered light is converted to Raman shifts, which represent the difference in energy between the original beam of light and the scattered light, usually measured in wavenumbers. Wavenumber is a unit that is defined as the number of wavelengths per unit distance typically reciprocal centimeters (cm−1) but can be interchangeable with the wavelength expressed in nm [10]. The Raman shift is employed to ensure that data can be easily compared, even when different laser wavelengths are utilized.

Line 86

It was corrected accordingly to the Reviewer’s suggestions.

Line 138

It was rephrased into: Raman effect generates the signal of low intensity and thus the spontaneous Raman spectroscopy is time-consuming.

We use RS as an abbreviation for spontaneous Raman scattering and SRS for stimulated Raman scattering.

Comments on the Quality of English Language

The highlighted parts of the introduction and the paragraph from section 3.1 were rewritten according to the suggestions of the Reviewer. Section 3.1.1 Hematopoietic stem cells was added to make sure that the RS applications in stem cell research have been clearly described.

Reviewer 2 Report

Comments and Suggestions for Authors

This manuscript is a review of the use of Raman spectra in Hematology and Hematooncology. This paper first introduces the mechanism of Raman spectroscopy, then the pipeline of data processing, finally application in medic. This work is well organized and well written, can be accepted after making following minor revision.

1.     In sector 3.2.2, line 323-324. The original paper made such conclusion under the temperature of 15-37°C, authors are supposed to be as accurate as them. As sample temperature below 15°C is possible and temperature may impact the Raman peaks.

2.     Authors introduced conventional data processing methods like PCA, cluster analysis, would authors please also introduce some latest methods like machine learning?

Author Response

  1. In sector 3.2.2, line 323-324. The original paper made such conclusion under the temperature of 15-37°C, authors are supposed to be as accurate as them. As sample temperature below 15°C is possible and temperature may impact the Raman peaks.

Line 320-322 was changed to: For example, the comparison between non-Hodgkin lymphoma (NHL) JMP-1/MCL and Hodgkin lymphoma MDA-V cell lines allowed for identification under different temperature as the measurements were taken in the range between 15-37°C.

  1. Authors introduced conventional data processing methods like PCA, cluster analysis, would authors please also introduce some latest methods like machine learning?

Some of the statistics-based machine learning algorithms were briefly noted in paragraph 119-125 but as requested, we expanded that section to highlight the importance of ML in modern spectroscopy analysis. Entire new paragraph have been added.

Reviewer 3 Report

Comments and Suggestions for Authors

The article discusses the use of Raman spectroscopy as a research and diagnostic tool in clinical hematology and hematooncology. However, the authors could improve the organization and description of the article. For instance, the review lacks graphical abstracts, images, or tables, which could aid in comparing and evaluating previous research and highlighting the article's main specific graph. Additionally, some of the references in the introduction are missing, which reduces the effectiveness of the article.

Author Response

References 1-8 were added to the introduction as suggested.

Graphical abstract was created. Two tables summarizing the utility of RS in hematological diagnostics were added to the section 3 to improve readability of the text.

Reviewer 4 Report

Comments and Suggestions for Authors

Comments are attached

Comments on the Quality of English Language

Minor editing of English language required

Author Response

Comment #1

The citation of the earliest Raman paper was introduced by the authors as a "honorary citation" - honoring the founder of the research field. To improve the quality of the text and readability, the text was rephrased as suggested linking theterm "promising" with the applications described in the text.

Comment #2

As suggested a table summarizing the utility of RS in hematological diagnostics was added at the end of section 3.

Round 2

Reviewer 1 Report

Comments and Suggestions for Authors

All my issues are properly addressed now.

Reviewer 3 Report

Comments and Suggestions for Authors

There is a well-revised version available from the authors. It would be appropriate to accept it. 

Reviewer 4 Report

Comments and Suggestions for Authors

My concern wasn't about the cited article, but about the word "promising" being used for a technology that's almost 100 years old. The word has been removed, but I don't think thorough bibliographic research has been done on all the applications of Raman spectroscopy in other fields. The introduction hasn't been expanded, and I don't consider the revision work done to be sufficient. In my opinion, the article cannot be published in its current form.